# Does Exposure of Lead and Cadmium Affect the Endometriosis?

**DOI:** 10.3390/ijerph18179077

**Published:** 2021-08-28

**Authors:** Min-Gi Kim, Young-Sun Min, Yeon-Soon Ahn

**Affiliations:** 1Department of Occupational and Environmental Medicine, Dankook University Hospital, College of Medicine, Dankook University, Cheonan-si 31116, Korea; searchthing@naver.com; 2Department of Occupational and Environmental Medicine, Soonchunhyang Cheonan Hospital, College of Medicine, Soon-chunhyang University, Cheonan-si 31151, Korea; mys0303@hanmail.net; 3Department of Preventive Medicine, College of Medicine, Yonsei University, Wonju 26426, Korea; 4Institute of Genomic Cohort, College of Medicine, Yonsei University, Wonju 26426, Korea

**Keywords:** lead, cadmium, endometriosis, synergic effect

## Abstract

This study aimed to investigate the effects of blood lead levels (BLLs) and lead and cadmium exposure on endometriosis (EM). The study cohort consisted of female workers who underwent a lead-associated special medical examination between 1 January 2000 and 31 December 2004 (*n* = 26,542). The standard admission rate (SAR) and admission odds ratio (OR) for EM were calculated using the general population and noise-exposed groups, respectively, for the same period as the reference standards. The SAR for EM was 1.24 (95% confidence interval (CI): 1.03–1.48) in lead-exposed workers and 1.44 (95% CI: 1.11–1.85) in workers with BLLs < 5 μg/dL when compared with that of the general population. Admission ORs of EM in lead-exposed workers and those with BLLs < 5 μg/dL were statistically higher than those of noise-exposed workers (OR, 1.40; 95% CI, 1.15–1.70 and OR, 1.48; 95% CI, 1.11–1.98, respectively). The relative excess risk due to interaction of lead and cadmium was 0.33. Lead exposure was associated with EM admission. EM admission in lead-exposed workers with BLLs < 5 μg/dL was statistically higher than that of the general population and noise-exposed workers. Co-exposure to lead and cadmium has a synergistic effect with EM.

## 1. Introduction

Endometriosis (EM) is a common, often chronic, inflammatory condition characterized by the presence of endometrium outside the uterus, mainly in the pelvic organs and tissues [1]. EM is also an oestrogen-dependent gynaecologic disease with lasting implications for some women’s fertility, somatic health, and overall quality of life. It is commonly associated with pelvic pain, menorrhagia, dyspareunia, and infertility [2]. The prevalence of endometriotic disease has shown to be approximately 5% of the reproductive age, with a peak between 25 and 35 years of age [2]. A 0.1% annual incidence of EM among women aged 15–49 years has been reported [3]. Missmer and Cramer reported that the prevalence of EM in asymptomatic women seeking sterilization is 2–18%; in those admitted with pelvic pain, it is 5–21%, whereas it is 5–50% in infertile women [4].

EM appears to be a disease with multiple causal factors; however, little is known about its exact aetiology [5]. Growing evidence suggests that endocrine-disrupting chemicals (EDCs) may be aetiologically involved in the development and severity of this disease [6].

To date, research focusing on EDCs and EM has largely focused on persistent chemicals, including dioxin-like compounds, organochlorine pesticides, polybrominated diphenyl ethers, polychlorinated biphenyls, perfluoroalkyl and polyfluoroalkyl substances, and select metals [5,7,8]. Many trace metals such as cadmium, lead, copper, chromium, and mercury have been shown to possess endocrine-disrupting properties in toxicological and epidemiological studies [5,7,8]. However, their role in EM has been inconsistent and limited [9].

Many investigators have shown lead- and cadmium-induced oxidative damage, which induces the formation of reactive oxygen species and lipid peroxidation and interferes with the antioxidant defence system, which includes the enzymes glutathione peroxidase, super-oxidase, and catalase [10,11,12]. Its role in EM remains largely unknown [10,11,12]. Impaired antioxidant defences can be a result of the inhibitory effects of lead on various enzymes, which in turn causes cells to be more susceptible to oxidative insult. The mechanism of lead-induced oxidative stress involves an imbalance between the generation and removal of reactive oxygen species in tissues and cellular components, causing damage to membranes, DNA, and proteins [11].

Epidemiological studies have mainly focused on cadmium, but few on lead and EM. Epidemiological studies on cadmium and EM have been conducted in Belgium, Japan, Sri Lanka, and the United States [5,9,12,13,14,15]. Of the six studies, only one, a cross-sectional nationally representative study in the United States based upon data from National Center for Health Statistics (NHANES, Hyattsville, MD, USA), found that cadmium was positively associated with EM diagnosis [9,16]. A recent study on Asian women found that the blood lead level (BLL) of EM was significantly higher than that in women without the disease [17]. This study aimed to investigate the relationship (1) between lead exposure and BLLs and EM admission and (2) between EM admission through co-exposure to lead and cadmium using a cohort of lead-exposed female workers and their hospitalization data. For a more obvious comparison of admission due to EM, multiple comparisons were performed using the external control group (general population) and internal control group (noise-exposed group).

## 2. Materials and Methods

### 2.1. Study Population and ASME

Since 2000, lead-associated special medical examination (ASME) data on exposure to occupational hazards (143 chemicals, 6 types of dust, 8 physical agents, and 19 metals, including lead) have been electronically stored and monitored by the Korean Occupational Safety and Health Agency (KOSHA) [18].

The ASME was conducted at more than 100 medical centres nationwide upon approval by KOSHA for the evaluation of Korean workers exposed to lead in the form of dust, fumes, etc. [18,19,20].

Interviews of personal information and health status performed by a physician were included in the specialized medical check-up for lead exposure. The workers’ personal information comprises their resident registration number (RRN, 13-digit unique number given to all Koreans) and employment history at the current workplace.

Physician interviews include the presence of lead-related signs and symptoms, whether workers wear a protective gear, and the duration of lead exposure

The workers underwent biological monitoring and clinical examination for lead and other co-exposures. The study cohort consisted of the KOSHA data of female workers exposed to lead and who had undergone a lead-associated medical check-up at least once between 1 January 2000 and 31 December 2004 (*n* = 26,542).

A total of 26,542 female workers underwent at least one ASME for lead, from 1 January 2000 to 31 December 2004. Their exposure to chemicals other than lead was classified using their work environment measurements and ASME data (cadmium, silica, and organic solvents).

To compare EM admission caused by lead exposure and BLLs in the lead-exposed group (study group), multiple comparisons were performed using an external control group (general population) and internal control group (noise-exposed group).

The external control group (general population) was randomly selected as 2% of the general Korean women population from the 2000 Korea National Health Insurance System (K-NHIS). The K-NHIS is a single-payer program with mandatory coverage for all Koreans [21].

Comparisons between lead and cadmium exposures were conducted only in the internal control group (noise-exposed group). This group is composed of female workers who underwent only noise-associated ASME in the same period as lead-exposed workers. To produce the same conditions with that of the study group, workers who underwent ASME for other metals (lead, cadmium, etc.) and organic solvents of the same duration were excluded.

### 2.2. Analysis of Blood Lead Levels

KOSHA conducts a semi-annual quality assurance program to monitor the analytic institutes and hospitals that participate in the ASME [20,22]. These institutes and hospitals should pass an analytic proficiency test to obtain a KOSHA license [20,23]. Only KOSHA-licensed institutes and hospitals could analyse BLLs of lead exposed workers in ASME, and this data is also reported to the KOSHA [20,22]. Analysis of BLLs are performed according to the KOSHA code H-09-1998 [20], which was based on the guideline of US National Institute for Occupational Safety and Health (NIOSH) [20,22,23,24,25].

Each KOSHA-licensed institute and hospital might have their own limit of detection (LOD) of BLL, but the information is not available [26]. The LOD of BLL was estimated at 0.85 μg/dL [26,27,28] and all results <0.85 μg/dL or reported as “not detected” in lead exposed workers were substituted using the equation $\sqrt{1/3\times 0.85}$ [27].

In each cohort, members’ BLLs varied from one to five, depending on how many times the worker underwent the ASME, including lead, during this period. The cumulative exposure level of exposed workers could not be identified; therefore, median BLLs were used as surrogates of cumulative exposure. BLLs of lead-exposed workers were classified as non-exposure (none: <5 μg/dL, ≥5 μg/dL). The “none” classification included female workers exposed to noise only and was used as a reference value in the analyses. A BLL of 5 μg/dL, the U.S. Centers for Disease Control and Prevention’s blood lead reference value, was used as the classification criterion for high exposure in this study [29]. In addition, according to the 2012–2014 Korean National Environmental Health Survey, the mean BLL for ages 19–64 years is 1.77 μg/dL, and the 95th percentile is 4.09 μg/dL [30].

### 2.3. Data on EM

EM-based morbidity was estimated from 2000 to 2005 using the Korea National Health Insurance Database (KNHICD). KNHICD data included personal RRN, admission and discharge dates, and disease diagnoses of patients. Each study participant’s RRN confirm their hospital admission for EM. Cases with a principal diagnosis of EM were included in the analysis; however, cases with a secondary diagnosis were excluded. Since 1989, the K-NHIS has covered all Korean residents [21], and nearly all hospitalized cases have been registered in the KNHICD. The diagnosis was based on the Korea Standard Classification of Disease and Cause of Death, fourth edition (KCD-4). Both the KCD-4 and International Classification of Disease, tenth revision (ICD-10), use the same classification for EM. The disease code analysed in this study was N80 (EM).

### 2.4. Statistical Methods

As EM is a chronic and progressive disease [1], patients are often hospitalized with the same illness. To indicate a single disease diagnosis, multiple admissions for the same disease were treated as a single admission. The first admission date was designated as the event date and was used to calculate the person-years. The person-years were measured using the factory enrolment data when workers used lead at work to the event date or the end of follow-up. To determine the differences in the occurrence of EM according to BLLs (<5 μg/dL, ≥5 μg/dL) in the study group, a chi-square test and *t*-test were performed. Analysis was performed according to the age (<30, 30–39, and ≥40 years), cadmium exposure (yes or no), silica exposure (yes or no), exposure to organic solvents (yes or no), and presence of EM (yes or no) (Table 1). For the external comparison (compared with the general population), the standard admission rates (SARs) with 95% confidence intervals (CIs) were estimated using the person-years and mortality computation software (Pancomp, version 11.0). The person-years of observation were jointly grouped according to age (20–24, 25–29, …, 65–70 years), two groups of blood levels (<5.0 μg/dL, ≥5.0 μg/dL), and two calendar years (2000–2002 and 2003–2005). Expected numbers of EM cases were calculated by multiplying the person-years with the age-, calendar-year-, and EM-specific incidence rates of the Korean general population, which permitted calculation of indirectly standardized ratios. The SAR was calculated by dividing the observed number of admissions in lead-exposed workers with the expected number of admissions in the general reference population (Table 2). A multiple logistic regression analysis was conducted to investigate the relationship between BLLs (<5 μg/dL, ≥5 μg/dL) (Table 3) and silica, organic solvent, lead, and cadmium exposure, lead–cadmium co-exposure, and EM development (Table 4) using the female noise-exposed worker group as reference. Collected data were analysed using SPSS version 23.0.

## 3. Results

### 3.1. Subsection Age, Tenure, and Cadmium, Silica, and Organic Solvent Exposures by BLLs in Lead-Exposed Female Workers

A total of 124 EM cases was observed. The mean age of those with BLLs ≥ 5 μg/dL were statistically higher than those with BLLs < 5 μg/dL. The proportion of cadmium and organic solvent exposures in those BLL ≥ 5 μg/dL was higher than those with BLL < 5 μg/dL. Silica exposure and the proportion of EM cases were not statistically significant between the two groups (Table 1).

### 3.2. The SARs of EM by BLLs (<5 μg/dL, ≥5 μg/dL) in Lead-Exposed Workers When Compared with That in the General Population

The total person-years of lead-exposed workers were 101,966. The SAR of in lead-exposed workers with EM was 1.24 (95% CI: 1.03–1.48), when compared to that of the general population. The SAR for EM in workers with BLLs < 5 μg/dL was 1.44 (95% CI: 1.11–1.85) when compared with that of the general population. No significant increase in the BLLs was observed in lead-exposed workers when compared to that of the general population. As the age increased by one year, the OR of EM significantly increased (OR, 1.03; 95% CI, 1.02–1.04) (Table 2).

### 3.3. The SARs of EM by BLLs (<5 μg/dL, ≥5 μg/dL) in Lead-Exposed Workers When Compared with That in the Noise Exposed Workers

The SAR for EM in workers with BLLs < 5 μg/dL was 1.48 (95% CI: 1.11–1.85) when compared with that of the noise exposed workers. No significant increase in the BLLs > 5 μg/dL was observed in lead-exposed workers when compared to that of noise exposed workers. As the age increased by one year, the OR of EM significantly increased (OR, 1.03; 95% CI, 1.02–1.04 (Table 3).

### 3.4. The Adjusted Odds of EM Caused by Cadmium and Lead Exposures and Their Co-Exposure and Silica and Organic Solvent Exposure

As the age increased by one year, the OR of EM significantly increased (OR, 1.03; 95% CI, 1.02–1.04). The ORs of EM in lead-exposed workers were statistically higher than those of noise-exposed workers (OR, 1.40; 95% CI, 1.15–1.70). The relative excess risk due to interaction (RERI) of lead and cadmium was calculated as follows: OR (lead and cadmium co-exposure) − OR (lead exposure, non-cadmium exposure) − OR (non-lead exposure and cadmium exposure) + 1 = 2.62 − 1.90 − 1.39 + 1 = 0.33 [31]. The synergistic effect of lead and cadmium on EM was observed [31] (Table 4).

## 4. Discussion

In this study, the SAR of BLLs (<5 μg/dL) in lead-exposed workers was significantly higher than that in the general population. BLLs < 5 μg/dL and lead exposure at work were associated with a significantly increased OR of HA in patients with EM after adjusting for age and exposure to cadmium, silica, and other organic solvents. While no statistical significance was observed at BLLs ≥ 5 µg/dL, co-exposure to cadmium and lead was associated with a significantly increased chance of hospital admission (HA) in patients with EM after adjusting for age and exposure to silica and other organic solvents.

Lead and cadmium are the most common environmental toxic pollutants that slowly accumulate in the body and mediate oxidative stress [11]. These two heavy metals mainly enter, accumulate, and remain in the body through the respiratory and digestive systems for a long period of time [32]. The half-lives of lead and cadmium are 5–20 years in the bones and 10–30 years in the kidneys, respectively [32]. Given the health hazards caused by these heavy metals, studies on lead or cadmium exposure and health impairment are ongoing [32].

According to the Agency for Toxic Substances and Disease Registry (2007), lead and cadmium (Cd) have serious health implications among heavy metals [33]. Among the many heavy metals listed in the d-orbital elements of the modern periodic table, cadmium and lead have gained prime importance due to their pathophysiological significance, as their bioaccumulation in living systems may cause severe damage to vital organs, that is, the reproductive systems, nervous system, gastrointestinal tract, and mucous tissues [33]. The Harlow Center for Biological Psychology has a cohort of rhesus monkeys that were exposed to low concentrations of lead acetate in utero or as infants [33]. Lead-exposed animals have been followed for 19 years and have developed four cases of inguinal hernia (males), three cases of EM (females), and one case of immunoblastic lymphoma (male). A strong association between EM and lead exposure was also observed (OR = 10.13, *p* < 0.001) [34].

Lai et al. reported that lead exposure leads to an imbalance of redox homeostasis, with occupationally exposed subjects showing higher levels of super-oxidative indicators and lower activities of anti-oxidative markers than that in controls, leading to lower superoxide dismutase activity, which is also associated with EM persistence [17]. The OR of women with BLLs in the third tertiles (>3 μg/dL) compared with those having a BLL in the first tertile (<0.38 μg/dL) was 2.44 (95% CI, 1.13–5.25), and the adjusted OR was 2.67 (95% CI, 1.17–6.07) in 224 Asian women with EM [17].

An association between lead exposure and EM was observed in the present study. However, it was only observed at a low concentration (<5 μg/dL). Although age was adjusted in the analyses, there were relatively more elderly people with high BLLs than in those with low BLLs. The mean age was also significantly higher in the high BLL group than those in the low BLL group. In the older age group, a low incidence of HA is expected in patients with EM due to the nature of the disease. In future studies, more detailed study results can be obtained if the study period and study subjects are increased.

A dose–response association between cadmium and EM was observed in a representative sample of US women who self-reported their EM status [9]. This association persisted in sub-analyses that were limited to women diagnosed in the past 10 years and women diagnosed since the last pregnancy [9]. However, the mean levels of cadmium in urine and blood did not differ between the 119 EM patients and 25 controls in another study [14].

In the Endometriosis: Natural History, Diagnosis and Outcomes (ENDO) Study, cadmium in the blood was associated with reduced odds of a diagnosis (adjusted OR = 0.55; 95% CI, 0.31–0.98) in 473 women aged 18–44 years and who were scheduled for laparoscopy/laparotomy when compared with 131 similarly aged women [15]. In this study, cadmium exposure was not statistically associated with HA in patients with EM [15]. The proportion of cadmium-exposed workers (0.1%) was very low among the lead-exposed female workers’ cohort in this study. To understand the relationship between cadmium exposure and EM, it is necessary to investigate the relationship between the level of cadmium in the blood and urine and EM using more study subjects exposed to cadmium.

The potential effect modification between lead and cadmium co-exposure can be assessed as a departure from additive effects. The RERI is considered the standard measure for interaction on an additive scale [33]. If it is modified to reflect the use of OR, the RERI is 033. This indicates that lead and cadmium have a synergistic association beyond an additive association. Lead and cadmium interaction in patients with EM was 0.13, and the synergic index was 1.25 [33].

Synergism typically occurs when at least one component affects the biological system. It occurs when the effect of the combination is greater than that predicted by the summed activity of each component individually at the same level of exposure that occurs in the mixture [30]. Living organisms are frequently exposed to a mixture of xenobiotics (heavy metals, pesticides, toxic gases, etc.) simultaneously [33]. Xenobiotic substances have been reported to cause toxicity in animals and key organs of humans [30]. Therefore, the combined interactions between xenobiotic substances and xenobiotic and animal systems are very important [33]. However, little is known about the biological effects of lead and cadmium co-exposure. This study is the first to suggest a synergistic effect of lead and cadmium on EM. To confirm this, more detailed animal and epidemiological studies are needed.

The limitations of this study are as follows: First, using only electronic data collected through ASME (secondary data), we could not investigate potential confounders related to EM, such as short menstrual cycle, family history of EM, genetic polymorphisms and smoking, lower body mass index, and lower parity [35]. The results of our study could be a possible finding due to residual or unmeasured confounding variables. Analysing smoking history in 2001, the current smoking rate of lead-exposed female workers (2.0%) was lower than that of Korean women (5.3% in 2001). Although the smoking rate in lead exposure cohort was not higher than general population, it is possible that other variables that the authors did not investigate might have influenced the outcomes. The second limitation was the lack of complete information on exposure history, including previous lead exposure prior to cohort construction, Third, the exposure correction of organic solvents related to EM, such as BPA and organic chlorine, was not individually corrected, but was collectively referred to as a single factor of exposure to an organic solvent. However, the result of each organic solvent correction was not different from those of the correction with an organic solvent exposure as a single factor. Finally, only HA data were used in this study. Due to the nature of EM, there might have been quite a few patients who only sought outpatient care without admission.

However, as both groups are covered by the same health insurance (NHIS), the impact of the limitation of admission data will not be significant [21]. Additionally, the authors have demonstrated the potential risk within certain age groups due to nature course of EM. The effects of lead and cadmium co-exposure on disease other than EM needs further investigation.

Long term follow-up of this study is required to determine the effect of lead on EM at BLLs ≥ 5 µg/dL and the effect of cadmium on EM. In particular, the number of female workers exposed to cadmium in this study were small (*n* = 277). Even in 2014, only 0.37% female workers (*n* = 126) exposed to cadmium among female workers manufacturing industry in South Korea which consisted of five of more workers [36]. Therefore, a multi-national study will be needed to determine the effect between cadmium and EM.

Despite these limitations, this study has several strengths. First, this the first study to describe the synergic association between lead and cadmium co-exposure and EM. Second, this study has a large sample size and consisted of a retrospective cohort design. Third, the data collection was carried out by a trustworthy organization. Finally, lead exposure, as well as BLLs (biological exposure index of lead) also associated with admission in patients with EM. Co exposure to cadmium and lead also associated with admission for EM in this study. In view of its association with EM, lead exposure of female workers should be reduced as much as possible. In particular, it is considered necessary to monitor BLLs and cadmium exposure indicators in the lead and cadmium co-exposed group.

## 5. Conclusions

Lead exposure was associated with admission in patients with EM through a 5-year follow-up of South Korean lead-exposed female workers. In addition, co-exposure to lead and cadmium was synergistically associated with admission in patients with EM. However, cadmium exposure itself was not statistically associated with HA in patients with EM.

Lead and cadmium exposure of female workers should be reduced as much as possible. In particular, it is considered necessary to monitor BLLs and cadmium exposure indicators in the lead and cadmium co-exposed group.

## Figures and Tables

**Table 1 ijerph-18-09077-t001:** Age, Tenure, Cadmium, Silica, and Organic Solvents Exposure by BLLs in Lead Exposed Female Workers.

Blood Lead Level		<5 μg/dL	≥5 μg/dL	*p*-Value
Variables		*n* (%)	*n* (%)	
		13,110 (49.4)	13,432 (50.6)	
Age (Years)	<30	8157 (62.2)	7119 (53.0)	0.00
	30–39	3375 (25.7)	4236 (31.5)	
	≥40	1578 (12.1)	2077(15.5)	
Mean (Age)		26.03 ± 10.45	27.78 ± 10.94	0.00
Cadmium	Yes	90 (0.7)	187 (1.4)	0.00
	No	13,019 (99.3)	13,246 (98.6)	
Silica	Yes	372 (2.8)	358 (2.7)	0.41
	No	12,737 (97.2)	13,075 (97.3)	
Organic solvent	Yes	5063 (38.6)	5533 (41.2)	0.00
	No	8046 (61.4)	7900 (58.8)	
Endometriosis (*n*)	Yes	63 (0.5)	61 (0.5)	0.79
	No	13,047 (99.5)	13,369 (99.5)	

**Table 2 ijerph-18-09077-t002:** The SARs of EM by BLLS (<5 μg/dL, ≥5 μg/dL) in Lead exposed workers when compared with general population.

Blood Lead Level	Total	<5 μg/dL	≥5 μg/dL
Person-year	101,966	46,905	55,060
Endometriosis cases	124	63	61
SAR (95% CI)	1.24 (1.03–1.48)	1.44 (1.11–1.85)	1.09 (0.83–1.39)

SAR: standard admission rate; CI: confidence interval.

**Table 3 ijerph-18-09077-t003:** The Adjusted Odds of EM by BLLs (<5 μg/dL, ≥5 μg/dL) and Cadmium, Silica, and Organic Solvents.

	OR ^1^	95% CI
Age	1.03	1.02–1.04
Cadmium	1.88	0.57–6.21
Silica	0.77	0.23–2.53
Organic Solvent	0.69	0.46–1.03
Lead (<5 μg/dL)	1.48	1.11–1.98
Lead (≥5 μg/dL)	1.31	0.97–1.76

^1^ Adjusted odds ratio (OR) by age (>30, 30–39, and ≥40), cadmium, silica, organic solvent, and lead (<5 μg/dL, ≥5 μg/dL).

**Table 4 ijerph-18-09077-t004:** The Adjusted Odds of EM by Cadmium and Lead exposure and their Co-Exposure, Silica, and Organic Solvents exposure.

	OR ^1^	95% CI
Age	1.03	1.02–1.04
Silica	0.71	0.26–1.93
Organic Solvent	0.68	0.49–0.95
Lead * Cadmium		
Lead	1.39	1.14–1.67
Cadmium	1.90	0.36–10.04
Lead and Cadmium	2.62	0.97–7.09

^1^ Adjusted odds ratio by age (>30, 30–39, and ≥40), silica, organic solvent, lead, cadmium and lead and cadmium.

## Data Availability

Not applicable.

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
