# Peer review of "Does Exposure of Lead and Cadmium Affect the Endometriosis?"

_ijerph, 2021, doi:10.3390/ijerph18179077_

Round 1

Reviewer 1 Report

This is a secondary data analysis trying to understand the risk of developing EM from lead and cadmium exposure. The study is with great impact, since EM is an estrogen-dependent disease and might be affected by endocrine-disruptors such as heavy metals. However, the main data from the manuscript has major statistic issues that might affect the outcome of the study. The manuscript also has several writing problems with the following comments that need to be improved or addressed:

  • Please spell out when using an acronym for the first time (EM in the abstract, and HA in line 211 in the discussion).
  • What does tenure in table 1 mean?
  • The results showed more harmful effect among those with BLLs <5ug/dL than those with BLLs >=5ug/dL is confusing. What’s the limit of detection (LOD) for BLLs? How many of the participants were below LOD in the <5ug/dL group?
  • The authors mentioned in the methods that the analysis was performed according to the age (<30, 30–39, 40–40, ≥50 years), but in Table 1, the stratification was only up to ≥40 years. Also typo in 40-40.
  • In Table 1, if testing using a chi-square test for the number of endometriosis (yes/no) between <5ug/dL and >=5ug/dL, the chi-square statistic is 0.099. The p-value is .753044. The result is not significant at p < .05. Please check the statistics again.
  • Line 287-289: “we do not think that there will be no significant difference in the smoking rate of lead-exposed workers compared to those of noise-exposed female workers.” Typo in the discussion?
  • Line 289: please check on typo and grammar. Also line 289 seems to be redundant with line 291.

Author Response

Response to Reviewer 1 Comments

Manuscript ID: IJERPH (ISSN 1660-4601)

<Thank you letter to reviewers>

We appreciate your critical review of our work and your suggestions for improving the quality of our manuscript. Based on the comments, we have provided point-by-point responses and have made the associated modifications to the manuscript.

Thank you in advance for your time and attention!

Point 1

Specific comments

Response 1

This is a secondary data analysis trying to understand the risk of developing EM from lead and cadmium exposure. The study is with great impact, since EM is an estrogen-dependent disease and might be affected by endocrine-disruptors such as heavy metals. However, the main data from the manuscript has major statistic issues that might affect the outcome of the study. The manuscript also has several writing problems with the following comments that need to be improved or addressed:

•     

Thank you for your valuable comments. .

•    What does tenure in table 1 mean?

Thank you for your valuable comments. .

Tenure is helpful resource for understanding for understanding lead exposure feamle workers. However, since no separate analysis was conducted about tenure, it was removed the table and contents.

•    The results showed more harmful effect among those with BLLs <5ug/dL than those with BLLs >=5ug/dL is confusing.

•     

Thank you for your valuable comments. .

The prevalence of endometriotic disease has shown to be approximately 5% of the reproductive age, with a peak between 25 and 35 years of age.

Howcer. This study focused the hospital admisson case. The ratio might be lower than outpatient cases.

An association between lead exposure and EM was observed in the present study. However, it was only observed at a low concentration (<5 μg/dL). Although age was adjusted in the analyses, there were relatively more elderly people with high BLLs than in those with low BLLs. The mean age was also significantly higher in the high BLL group than those in the low BLL group. In the older age group, a low incidence of HA is expected in patients with EM due to the nature of the disease. In future studies, more detailed study results can be obtained if the study period and study subjects are increased

Blood Lead Level

< 5ug/dl

≥ 5ug/dl

p-value

Variables

n(%)

n(%)

13,110(49.4)

13,432 (50.6)

Age (Years)

< 30

8,157 (62.2)

7,119 (53.0)

0.00

30-39

3,375(25.7)

4,236 (31.5)

≥ 40

1,578(12.1)

2,077(15.5)

Mean(Age)

Mean(age)

26.03±10.45

27.78±10.94

0.00

•    What’s the limit of detection (LOD) for BLLs? How many of the participants were below LOD in the <5ug/dL group?

Thank you for your valuable comments.

.

KOSHA-licensed institutes and hospitals might have its level of a limit of detection of BLL; however, the information is not available. [23]

For measuring BLL, a standard analytical method of the KOSHA (KOSHA GUIDE H-21-2011) was employed, which is largely based on the US NIOSH method using graphite furnace atomic absorption spectrophotometry [23-24]. The LOD of BLL was estimated at 0.85 μg/dL[23-24]

all results <0.85 μg/dL or reported as “not detected” in lead exposed workers were substituted using the equation 1 / 3 × 0.85.

The following are the changes to the text.

Sampling and analysis of BLLs are performed in accordance with the KOSHA code H-09-1998 [20, 22], which was developed by KOSHA based on guidelines established by the National Institute for Occupational Safety and Health [22-23].

KOSHA-licensed institutes and hospitals might have its level of a limit of detection of BLL; however, the information is not available [23]. For measuring BLL, a standard analytical method of the KOSHA (KOSHA GUIDE H-21-2011) was employed, which is largely based on the US NIOSH method using graphite furnace atomic absorption spectrophotometry [23]. The LOD of BLL was estimated at 0.85 μg/dL.[23-24]. The limit of detection was set at 0.85 μg/dL [7], and all results <0.85 μg/dL or reported as “not detected” in lead exposed workers were substituted using the equation 1 / 3 × 0.85.

Dong-Hee Koh, Ju-Hyun Park

[23] Dong-Hee Koh, Ju-Hyun Park, Sang-Gil Lee, Hwan-Cheol Kim, Hyejung Jung, Inah Kim, Sangjun Choi, Donguk Park, Estimation of Lead Exposure Intensity by Industry Using Nationwide Exposure Databases in Korea, Safety and Health at Work, 2021,

[24] Kim JH, Kim EA, Koh DH, Byun K, Ryu HW, Lee SG. Blood lead levels of Korean lead workers in 2003-2011. Ann Occup Environ Med. 2014 Oct 1;26:30. doi: 10.1186/s40557-014-0030-3. PMID: 25379187; PMCID: PMC4209518.

•    The authors mentioned in the methods that the analysis was performed according to the age (<30, 30–39, 40–40, ≥50 years), but in Table 1, the stratification was only up to ≥40 years. Also typo in 40-40.

•     

Thank you for your valuable comments.

The auhors changed it like this..

age (<30, 30–39, ≥40 years),

•    In Table 1, if testing using a chi-square test for the number of endometriosis (yes/no) between <5ug/dL and >=5ug/dL, the chi-square statistic is 0.099. The p-value is .753044. The result is not significant at p < .05. Please check the statistics again.

•     

Thank you for your valuable comments.

The auhors change it like this.

Blood Lead Level

< 5ug/dl

≥ 5ug/dl

p-value

Variables

n(%)

n(%)

13,110(49.4)

13,432 (50.6)

Endometriosis (n)

Yes

63 (0.5)

61 (0.5)

0.79

No

13,047 (99.5)

13,369 (99.5)

•    Line 287-289: “we do not think that there will be no significant difference in the smoking rate of lead-exposed workers compared to those of noise-exposed female workers.” Typo in the discussion?

•     

Thank you for your valuable comments.

The authors changed it like this.

Although, the smoking rate in lead exposure cohort was not higher than general population, it is possible that other variables that the authors did not investigate might have influenced the outcomes

•    Please spell out when using an acronym for the first time (EM in the abstract, and HA in line 211 in the discussion).

•     

Thank you for your valuable comments.

hospital admission (HA)

•    Line 289: please check on typo and grammar. Also line 289 seems to be redundant with line 291.

•     

Thank you for your valuable comments.

The authors removed the text 289-291 as you pointed out.

Reviewer 2 Report

The paper proposed for publication is structured in 5 chapters. The first chapter, the introductory one, presents the motivation for choosing this research topic.

The second chapter, entitled Materials and Methods, begins with the Description of the Study Population and ASME, continues with the Sampling and analysis of blood lead levels, Data on EM and Statistical methods. 

The third chapter, is dedicated to the results, presented in tabular form. 

Please address small shortcuts related to the  table 1 (Age (Years) ≥ 40  - ≥ 5ug/dl, may be need "," in 2077(56.8) ?)

The fourth chapter presents the discussions on the researched topic.

The fifth chapter highlights the research results

The paper has a high scientific character, presents novelty elements for the chosen topic and I consider that it meets all the conditions to be published.

Author Response

Response to Reviewer 2 Comments

Manuscript ID: IJERPH (ISSN 1660-4601)

<Thank you letter to reviewers>

We appreciate your critical review of our work and your suggestions for improving the quality of our manuscript. Based on the comments, we have provided point-by-point responses and have made the associated modifications to the manuscript.

Thank you in advance for your time and attention!

Specific comments

Response

The paper proposed for publication is structured in 5 chapters. The first chapter, the introductory one, presents the motivation for choosing this research topic.

The second chapter, entitled Materials and Methods, begins with the Description of the Study Population and ASME, continues with the Sampling and analysis of blood lead levels, Data on EM and Statistical methods. 

The third chapter, is dedicated to the results, presented in tabular form. 

Please address small shortcuts related to the  table 1 (Age (Years) ≥ 40  - ≥ 5ug/dl, may be need "," in 2077(56.8) ?)

The fourth chapter presents the discussions on the researched topic.

The fifth chapter highlights the research results

The paper has a high scientific character, presents novelty elements for the chosen topic and I consider that it meets all the conditions to be published.

Thank you for your valuable comments.

The authors have made corrections as you pointed out.

Blood Lead Level

< 5ug/dl

≥ 5ug/dl

p-value

Variables

n(%)

n(%)

13,110(49.4)

13,432 (50.6)

Age (Years)

< 30

8,157 (62.2)

7,119 (53.0)

0.00

30-39

3,375(25.7)

4,236 (31.5)

≥ 40

1,578(12.1)

2,077(15.5)

Mean(Age)

Mean(age)

26.03±10.45

27.78±10.94

0.00

In addtion to, Table 1 has been modified as follows, and a description of the blood lead concentration has been added.

Table 1. Age, Tenure, Cadmium, Silica and Organic Solvents Exposure by BLLs in Lead Exposed Female Workers.

Blood Lead Level

< 5ug/dl

≥ 5ug/dl

p-value

Variables

n(%)

n(%)

13,110(49.4)

13,432 (50.6)

Age (Years)

< 30

8,157 (62.2)

7,119 (53.0)

0.00

30-39

3,375(25.7)

4,236 (31.5)

≥ 40

1,578(12.1)

2,077(15.5)

Mean(Age)

26.03±10.45

27.78±10.94

0.00

Cadmium

Yes

90 (0.7)

187 (1.4)

0.00

No

13,019 (99.3)

13,246 (98.6)

Silica

Yes

372 (2.8)

358 (2.7)

0.41

No

12,737 (97.2)

13,075 (97.3)

Organic solvent

Yes

5,063 (38.6)

5,533 (41.2)

0.00

No

8.046 (61.4)

7,900 (58.8)

Endometriosis (n)

Yes

63 (0.5)

61 (0.5)

0.79

No

13,047 (99.5)

13,369 (99.5)

2.2. Sampling and analysis of blood lead levels

. KOSHA-licensed institutes and hospitals might have its level of a limit of detection of BLL; however, the information is not available. For measuring BLL, a standard analytical method of the KOSHA (KOSHA GUIDE H-21-2011) was employed, which is largely based on the US NIOSH method using graphite furnace atomic absorption spectrophotometry [24]. The LOD of BLL was estimated at 0.85 μg/dL [24-25]. The limit of detection was set at 0.85 μg/dL [24-25]and all results <0.85 μg/dL or reported as “not detected” in lead exposed workers were substituted using the equation 1 / 3 × 0.85 [25].

Reviewer 3 Report

The authors present a study in which the association of lead and cadmium exposure with the emergence of endometriosis was investigated using data taken from an information system. Although the authors present references in the text that support the possibility of metal poisoning being associated with endometriosis, I consider that the small number of cases of women with endometriosis (124) in the study and the statistical analyzes chosen by the authors do not allow the results found to support the conclusions of the study.

Author Response

Response to Reviewer 3 Comments

Manuscript ID: IJERPH (ISSN 1660-4601)

<Thank you letter to reviewers>

We appreciate your critical review of our work and your suggestions for improving the quality of our manuscript. Based on the comments, we have provided point-by-point responses and have made the associated modifications to the manuscript.

Thank you in advance for your time and attention!

Specific comments

Response

The authors present a study in which the association of lead and cadmium exposure with the emergence of endometriosis was investigated using data taken from an information system. Although the authors present references in the text that support the possibility of metal poisoning being associated with endometriosis, I consider that the small number of cases of women with endometriosis (124) in the study and the statistical analyzes chosen by the authors do not allow the results found to support the conclusions of the study.

Thank you for your valuable comments. .

As pointed out, a small number of endometriosis cases were observed in this study..

In this study, the number of patients was small because only inpatients were included among em cases..

In South Korea, lead exposed female workrs mihgt have relatively poor working conditions and economic conditions than the general population and general workers. Therefore, it might affect it (small cases)

Through this study, the authors believe that there is a need for longer follow-up studies and studies in other countries with larger subjects.

Through it, the effects of lead and cadmium on endometriosis are likely to be more clearly defined.

Reviewer 4 Report

The manuscript reports a study regarding the correlación between the exposure to lead and cadmium and endometriosis. The leves of the two toxic elements in the blood were extracted from existing medical exam récords of >20,000 subjects collected for five years. Although there is no direct evidence to show that long term exposure to lead and/or cadmium could lead to endometriosis, one of the limitations of the study, the authors have demonstrated the potential risk within certain age groups. The study design, data collection and interpretation are well described. It might be worth exploring/comparing individual effect of lead and cadmium on the female subjects. Additionally, it might be helpful to specifiy whether the co-exposure to lead and cadmium could trigger additive or syngentic effects.

Author Response

Response to Reviewer 4 Comments

Manuscript ID: IJERPH (ISSN 1660-4601)

<Thank you letter to reviewers>

We appreciate your critical review of our work and your suggestions for improving the quality of our manuscript. Based on the comments, we have provided point-by-point responses and have made the associated modifications to the manuscript.

Thank you in advance for your time and attention!

Specific comments

Response

The manuscript reports a study regarding the correlación between the exposure to lead and cadmium and endometriosis. The leves of the two toxic elements in the blood were extracted from existing medical exam récords of >20,000 subjects collected for five years. Although there is no direct evidence to show that long term exposure to lead and/or cadmium could lead to endometriosis,. The study design, data collection and interpretation are well described. It might be worth exploring/comparing individual effect of lead and cadmium on the female subjects. Additionally, it might be helpful to specifiy whether the co-exposure to lead and cadmium could trigger additive or syngentic effects..

Thank you for your valuable comments. .

Tha authors revised the text as you pointed out.

one of the limitations of the study, the authors have demonstrated the potential risk within certain age groups

Thank you for your valuable comments.

However, since both groups are covered by the same health insurance (NHIS), the impact of the limitation of admssion data will not be significant [21]. Additionally, the authors have demonstrated the potential risk within certain age groups due to nature course of EM. The effects of lead and cadmium co-exposure on disease other than EM needs further investigation.

. It might be worth exploring/comparing individual effect of lead and cadmium on the female subjects

Thank you for your valuable comments.

Lead exposure was associated with admission in patients with EM through a 5-year follow-up of South Korean lead-exposed female workers. In addition, co-exposure to lead and cadmium was synergistically associated with admission in patients with EM. However, cadmium exposure itself was not statistically associated with HA in patients with EM

Additionally, it might be helpful to specifiy whether the co-exposure to lead and cadmium could trigger additive or syngentic effects..

Thank you for your valuable comments.

Despite these limitations, this study has several strengths. First, this a first study to describe the synergic association with between lead and cadmium co-exposure and EM. Second, this study has the large sample size and consisted of a retrospective cohort design.

Lead and cadmium exposure of female workers should be reduced as much as possible. In particular, it is considered necessary to monitor BLLs and cadmium exposure indicators in the lead and cadmium co-exposed group

Round 2

Reviewer 1 Report

Many typos were found from the authors' response. There were also issues in plagiarism from revised sentences:

Sampling and analysis of BLLs are performed in accordance with the KOSHA code H-09-1998 [20, 22], which was developed by KOSHA based on guidelines established by the National Institute for Occupational Safety and Health [22-23].

Source: J Occup Health. 2020 Jan-Dec; 62(1): e12107.

For measuring BLL, a standard analytical method of the KOSHA (KOSHA GUIDE H-21-2011) was employed, which is largely based on the US NIOSH method using graphite furnace atomic absorption spectrophotometry [23].

Source: Safety and Health at Work. 2021 July 17. In Press.

Author Response

Response to Reviewer 1 Comments

Manuscript ID: IJERPH (ISSN 1660-4601)

<Thank you letter to reviewers>

We appreciate your critical review of our work and your suggestions for improving the quality of our manuscript. Based on the comments, we have provided point-by-point responses and have made the associated modifications to the manuscript.

Thank you in advance for your time and attention!

Point 1

Specific comments

Response 1

Many typos were found from the authors' response.

Thank you for your valuable comments.

The authors apolgize for many typos in author’s response.  

There were also issues in plagiarism from revised sentences:

Sampling and analysis of BLLs are performed in accordance with the KOSHA code H-09-1998 [20, 22], which was developed by KOSHA based on guidelines established by the National Institute for Occupational Safety and Health [22-23].

Source: J Occup Health. 2020 Jan-Dec; 62(1): e12107.

Thank you for your valuable comments

The authors corrected the text as you pointed out. The thesis you presented is reference 23.

Analysis of BLLs are performed according to the KOSHA code H-09-1998 [20], which was based on the guideline of US National Institute for Occupational Safety and Health (NIOSH) [20,22-25].

20.Kim KR, Lee SW, Paik NW, Choi KG. Low-level lead exposure among South Korean lead workers, and estimates of associated risk of cardiovascular disease. J Occupat Environ Hyg. 2008; 5(6).

21. Song SO, Jung CH, Song YD, Park CY, Kwon HS, Cha BS, et al. Background and data configuration process of a nationwide population-based study using the Korean national health insurance system. Diabetes Metab J. 2014;38:395-403.

22.Min YS, Ahn YS. The association between blood lead levels and cardiovascular diseases among lead-exposed male workers. Scand J Work Environ Health. 2017;43(4):385-390.

23.Kim MG, Kim YW, Ahn YS. Does low lead exposure affect blood pressure and hypertension? J Occup Health. 2020 Jan;62(1):e12107

24.Korean Occupational Safety and Health Administration (KOSHA). An analytical guidance of biomarkers for Lead KOSHA CODE H-09-1998. Incheon, South Korea: KOSHA, 1998. (In Korean)

25. National Institute for Occupational Safety and Health (NIOSH). Method 7105 LEAD by GFAAS and 7082 LEAD by FAAS. In: PC Schlecht, PF O'Connor, editors. NIOSH Manual of Analytical Methods (NMAM) DHHS (NIOSH) Pub. no. 94-113, 4th ed. Cincinnati, OH: NIOSH; 1994

For measuring BLL, a standard analytical method of the KOSHA (KOSHA GUIDE H-21-2011) was employed, which is largely based on the US NIOSH method using graphite furnace atomic absorption spectrophotometry [23].

Source: Safety and Health at Work. 2021 July 17. In Press.

•     

Thank you for your valuable comments

The authors corrected the text as you pointed out. The thesis you presented is reference 26.

Analysis of BLLs are performed according to the KOSHA code H-09-1998 [20], which was based on the guideline of US National Institute for Occupational Safety and Health (NIOSH) [20,22-25].

Each KOSHA-licensed institute and hospital might have their own limit of detection (LOD) of BLL, but the information is not available [26]. The LOD of BLL was estimated at 0.85 μg/dL [26-28] and all results <0.85 μg/dL or reported as “not detected” in lead exposed workers were substituted using the equation  [27].

20.Kim KR, Lee SW, Paik NW, Choi KG. Low-level lead exposure among South Korean lead workers, and estimates of associated risk of cardiovascular disease. J Occupat Environ Hyg. 2008; 5(6).

21. Song SO, Jung CH, Song YD, Park CY, Kwon HS, Cha BS, et al. Background and data configuration process of a nationwide population-based study using the Korean national health insurance system. Diabetes Metab J. 2014;38:395-403.

22.Min YS, Ahn YS. The association between blood lead levels and cardiovascular diseases among lead-exposed male workers. Scand J Work Environ Health. 2017;43(4):385-390.

23.Kim MG, Kim YW, Ahn YS. Does low lead exposure affect blood pressure and hypertension? J Occup Health. 2020 Jan;62(1):e12107

24.Korean Occupational Safety and Health Administration (KOSHA). An analytical guidance of biomarkers for Lead KOSHA CODE H-09-1998. Incheon, South Korea: KOSHA, 1998. (In Korean)

25. National Institute for Occupational Safety and Health (NIOSH). Method 7105 LEAD by GFAAS and 7082 LEAD by FAAS. In: PC Schlecht, PF O'Connor, editors. NIOSH Manual of Analytical Methods (NMAM) DHHS (NIOSH) Pub. no. 94-113, 4th ed. Cincinnati, OH: NIOSH; 1994.

26.Koh,DH Park JH, Lee SG, Kim HC, Jung H, Kim I, Choi S, Park D, Estimation of Lead Exposure Intensity by Industry Using Nationwide Exposure Databases in Korea, Saf Health Work, 2021. In press.

27.Kim JH, Kim EA, Koh DH, Byun K, Ryu HW, Lee SG. Blood lead levels of Korean lead workers in 2003-2011. Ann Occup          Environ Med. 2014 Oct 1;26:30.

28.Occupational Safety and Health Research Institute: Standard of Biological Exposure Indices and Analytical Methods III. Incheon: 2010 (In Korean).

Reviewer 3 Report

None

Author Response

The authors present a study in which the association of lead and cadmium exposure with the emergence of endometriosis was investigated using data taken from an information system. Although the authors present references in the text that support the possibility of metal poisoning being associated with endometriosis, I consider that the small number of cases of women with endometriosis (124) in the study and the statistical analyzes chosen by the authors do not allow the results found to support the conclusions of the study.

Thank you for your valuable comments. .

As pointed out, a small number of endometriosis  cases were observed in this study..

In this study, the number of patients was small because only inpatients were included among em cases..

In South Korea, lead exposed female workrs mihgt have relatively poor working conditions and economic conditions than the general population.Therefore, it might affect it (small cases)

Through this study, the authors believe that there is a need for longer follow-up studies and studies in other countries with larger subjects.

Through it, the effects of lead and cadmium on endometriosis are likely to be more clearly defined.